# Weekly Oral Tenofovir Alafenamide Protects Macaques from Vaginal and Rectal Simian HIV Infection

**DOI:** 10.3390/pharmaceutics16030384

**Published:** 2024-03-11

**Authors:** Ivana Massud, Kenji Nishiura, Susan Ruone, Angela Holder, Chuong Dinh, Jonathan Lipscomb, James Mitchell, George M. Khalil, Walid Heneine, J. Gerardo Garcίa-Lerma, Charles W. Dobard

**Affiliations:** 1Laboratory Branch, Division of HIV Prevention, Centers for Disease Control and Prevention, Atlanta, GA 30333, USA; vve9@cdc.gov (I.M.); knishiur@gmail.com (K.N.); fyh1@cdc.gov (S.R.); fka5@cdc.gov (A.H.); vul9@cdc.gov (C.D.); eyk1@cdc.gov (J.L.); zjl9@cdc.gov (J.M.); wmh2@cdc.gov (W.H.); jng5@cdc.gov (J.G.G.-L.); 2Quantitative Sciences and Data Management Branch, Division of HIV Prevention, National Center for HIV, Viral Hepatitis, STD, and TB Prevention, Centers for Disease Control and Prevention, Atlanta, GA 30329, USA; uwm4@cdc.gov

**Keywords:** tenofovir alafenamide fumarate (TAF), tenofovir diphosphate (TFV-DP), non-human primate (NHP), HIV prevention, long-acting pre-exposure prophylaxis (LA PrEP), pharmacokinetics (PK)

## Abstract

Pre-exposure prophylaxis (PrEP) with a weekly oral regimen of antiretroviral drugs could be a suitable preventative option for individuals who struggle with daily PrEP or prefer not to use long-acting injectables. We assessed in macaques the efficacy of weekly oral tenofovir alafenamide (TAF) at doses of 13.7 or 27.4 mg/kg. Macaques received weekly oral TAF for six weeks and were exposed twice-weekly to SHIV vaginally or rectally on day 3 and 6 after each dose. Median TFV-DP levels in PBMCs following the 13.7 mg/kg dose were 3110 and 1137 fmols/10^6^ cells on day 3 and 6, respectively. With the 27.4 mg/kg dose, TFV-DP levels were increased (~2-fold) on day 3 and 6 (6095 and 3290 fmols/10^6^ cells, respectively). Both TAF doses (13.7 and 27.4 mg/kg) conferred high efficacy (94.1% and 93.9%, respectively) against vaginal SHIV infection. Efficacy of the 27.4 mg/kg dose against rectal SHIV infection was 80.7%. We estimate that macaque doses of 13.7 and 27.4 mg/kg are equivalent to approximately 230 and 450 mg of TAF in humans, respectively. Our findings demonstrate the effectiveness of a weekly oral PrEP regimen and suggest that a clinically achievable oral TAF dose could be a promising option for non-daily PrEP.

## 1. Introduction

Pre-exposure prophylaxis (PrEP) with daily oral regimens is highly effective in preventing HIV-1 infection and, as of 2023, more than 4.5 million individuals worldwide were receiving PrEP (https://avac.org/resource/infographic/cumulative-number-of-prep-initiations-for-q1-of-2023) (accessed on 11 October 2023) [1]. However, the clinical and public benefit of oral PrEP has been hindered by suboptimal adherence and high rates of premature PrEP discontinuation [2,3,4,5]. PrEP regimens that do not require daily dosing provide a desirable option for persons who are unable to adhere to daily dosing. In 2021, the US Food and Drug Administration (FDA) approved a bimonthly injection of the integrase inhibitor cabotegravir (CAB LA) as the first long-acting PrEP regimen. Despite an overall preference for injectable PrEP among end-users, studies also highlighted the need for a wide range of PrEP options that allow end-users to choose the PrEP modality that better suits their needs [6,7,8,9]. 

Although many factors affect adherence to prescribed medications, some studies have shown that compliance to oral medications is generally inversely proportional to dosing frequency [4,5,10,11]. For HIV prevention, fixed non-daily oral PrEP regimens are currently not available. Oral PrEP regimens consisting of antiretroviral (ARV) drugs with a long half-life may have the potential for weekly or monthly administration and offer advantages over parenteral drug delivery, including self-administration, convenience, and potential for lower cost [6,12,13]. The nucleoside reverse transcriptase translocator inhibitor Islatravir (ISL) was the first antiretroviral considered for long-acting oral PrEP due to its high potency and long intracellular half-life [14,15]. In non-human primates, a weekly oral dose of ISL conferred full protection against rectal infection with simian HIV (SHIV). These studies also identified a concentration of ISL-triphosphate (ISL-TP) of ~50 pmols per million peripheral blood mononuclear cells (PBMCs) needed for rectal protection [16]. In humans, oral doses of 60 and 120 mg of ISL maintained ISL-TP concentrations above the PrEP protection benchmark for at least four months. Unfortunately, clinical trials with monthly ISL were stopped due to unexpected decreases in total lymphocytes and CD4+ T-cell counts [17,18,19].

Tenofovir alafenamide (TAF), a prodrug of tenofovir, is a nucleotide reverse transcriptase inhibitor that has many desirable characteristics for long-acting oral PrEP. The pharmacologically active metabolite of TAF (tenofovir diphosphate, TFV-DP) has a long intracellular half-life in PBMCs [20,21,22]. In addition, TAF doses lymphoid tissues more efficiently than the previously approved tenofovir prodrug, tenofovir disoproxil fumarate (TDF) [23,24,25,26]. In humans, a daily 25 mg dose of TAF achieves 4- to 7-fold higher intracellular TFV-DP concentrations in PBMCs than a daily 300 mg TDF dose and reduces plasma TFV concentrations by ~90% [25,27,28,29]. Because of these pharmacologic advantages and low oral daily dose, we reasoned that a higher TAF dose may be suitable for a weekly regimen. In a small clinical trial, no serious adverse events were noted in groups of study participants that received daily oral TAF during 10 consecutive days at doses of 200 or 600 mg, which is eight and 24 times the clinical daily dose, respectively (NCT03739866).

To explore the potential of a weekly oral TAF regimen for HIV PrEP in humans, we investigated the pharmacokinetics (PK) and PrEP efficacy of a weekly oral dose of TAF in a macaque model. We tested two different doses of oral TAF (13.7 or 27.4 mg/kg) in validated macaque models of repeated rectal and vaginal SHIV exposures that predicted the clinical efficacy of daily oral emtricitabine (FTC) with TDF or TAF [30]. The doses of TAF used in our study were selected to maintain TFV-DP levels above the levels associated with vaginal PrEP protection established from a subcutaneous TAF implant [31]. 

## 2. Materials and Methods

### 2.1. Humane Care Guidelines

All animal protocols were approved by the Centers for Disease Control and Prevention (CDC) Institutional Animal Care and Use Committee. Pigtailed macaques were cared for by CDC veterinarians in compliance with the Guide for the Care and Use of Laboratory Animals, 8th Ed [32].

### 2.2. Drug Preparation and Dosing

Tenofovir alafenamide fumarate (TAF) was kindly provided by Laurus Labs. TAF powder was dissolved in 0.01 M phosphate-buffered saline (PBS) [pH 7.2] and administrered to anesthetized macaques by oral gavage.

### 2.3. Virus Challenge Stock

SHIV_SF162P3_ (SIVmac_239_ backbone with an HIV-1 subtype B, CCR5–tropic envelope) was obtained from the National Institutes of Health (NIH) AIDS Research and Reference Reagent Program and was propagated and then titrated in PBMCs from pigtailed macaques to generate a master stock (SHIV162p3 [4660 TCID_50_]). The master stock used for vaginal and rectal challenges was diluted to 1:20 and stored individually as 1 mL aliquots in liquid nitrogen and thawed prior to each challenge.

### 2.4. PK Assessments

We performed dose-ranging studies in female pigtailed macaques to identify a weekly oral TAF dose that would maintain TFV-DP levels in PBMCs above the protective benchmark (1100 fmol/10^6^ cells) for at least one week. This benchmark has been associated with high vaginal protection from a subcutaneous implant delivering TAF [31]. Macaques received a single dose of oral TAF at 13.7 mg/kg (*n* = 5) or 27.4 mg/kg (*n* = 2), and blood was collected over time (5, 72, 144, or 168 h post-dosing). Based on previous studies showing that a daily TAF dose of 1.5 mg/kg in macaques is equivalent to the clinical daily 25 mg TAF dose in humans, the weekly 13.7 and 27.4 mg/kg doses are estimated to be nine and 18 times the daily dose, respectively.

### 2.5. Vaginal Efficacy of Weekly Oral TAF

The efficacy of weekly oral TAF was investigated in an established pigtailed macaque model of repeated vaginal SHIV exposures [31]. In the first study group (Group 1), six macaques received 27.4 mg/kg of TAF once a week and were exposed vaginally to 1 mL of SHIV_162P3_ on days 3 and 6. This schedule was repeated for six consecutive weeks (12 total SHIV exposures) or until an animal became infected. Blood was collected at the time of the SHIV challenges (days 3 and 6 post-dose) to monitor for SHIV infection status and TFV-DP levels in PBMCs. The same experimental design and procedures were followed in a second group of macaques (Group 2), which received a lower weekly TAF dose of 13.7 mg/kg). Infection outcomes were compared to six untreated macaques challenged with the same SHIV inoculum twice per week (two real-time and four contemporary controls). Animals were considered protected if they remained seronegative for SHIV antibodies and negative for SHIV RNA during the six-week challenge period and the following nine weeks of drug washout. SHIV RNA in plasma was quantified by an RT-PCR assay with a sensitivity of 50 RNA copies/mL [33,34]. Virus-specific serologic responses were measured using a synthetic peptide enzyme immunoassay assay (BioRad, Genetic Systems HIV-1/HIV-2, Redmond, WA, USA).

### 2.6. Rectal Efficacy of Weekly TAF

After a drug washout and rest period of 16 weeks, five uninfected macaques from Group 1 and one additional naive macaque were enrolled in a study to evaluate rectal efficacy with the 27.4 mg/kg weekly TAF dose. Macaques were challenged twice weekly with the same SHIV dose at the same time (day 3 and 6) post-dosing Blood was collected 30 min before each SHIV challenge to monitor for SHIV infection status and TFV-DP levels in PBMCs. Infection outcomes were compared to two untreated macaques challenged rectally twice per week with the same SHIV inoculum and six historical controls challenged rectally, as previously described [35]. 

### 2.7. Analysis of TFV and TFV-DP Concentrations

TFV levels in plasmas were measured by high-performance liquid chromatography-tandem mass spectrometry (HPLC-MS/MS) (Sciex, Foster City, CA, Shimadzu Scientific, Columbus, MD, USA) [36]. Briefly, a cocktail of 500 µL of methanol containing the internal standard, isotopically labeled tenofovir (Toronto Research Chemicals Inc., Toronto, Canada), was added to 100 µL of plasma to precipitate proteins. After a brief centrifugation and removal of protein precipitates, the supernatant was evaporated to near dryness and re-suspended in 100 µL of mobile phase A (0.2% formic acid in water). Ten µL of the final solution was injected onto a UK-C18 column (100 × 1 mm, Imtakt, Portland, OR, USA) connected to an HPLC-MS/MS system. An aqueous–acetonitrile mobile-phase gradient was used to elute the drugs from the column and into the analyzer. Mass transitions used for tenofovir are 288.0/176.3 and 288.0/159.1 (*m*/*z*) and were monitored in positive mode. The drug concentrations were estimated from a standard curve with a range of 1–3000 ng/mL using Analyst software 1.7.3 (Sciex, Foster City, CA, USA) and a lower limit of quantification of 10 ng/mL.

TFV-DP was measured in PBMCs, as described previously [37]. TFV-DP concentrations were measured with an automated online weak anion-exchange solid-phase extraction method coupled with ion-pair chromatography-MS/MS. Briefly, TFV-DP was extracted from PBMCs (~3 million) by protein precipitation with 500 µL of 80% methanol containing 1000 ng of 13C-labeled TFV-DP as internal standards. Samples were extracted overnight at −80 °C, centrifuged, and dried to completion. Samples were then reconstituted in 100 µL of 50 mM ammonium acetate buffer (pH 7.0) and centrifuged at 17,000× *g* to remove insoluble particulates. The liquid chromatography method employed the use of a column-switching system (a 6-port valve and a 10-port valve), with four single-solvent line pumps and an autosampler (Shimadzu). The eluents are Pump A: 0.2% acetic acid, Pump B: 1% ammonium hydroxide, Pump C: 10 mM ammonium formate/30% acetonitrile/0.1% dimethylhexylamine (DMHA). The columns used were: WAX: Biobasic AX 5 um 10 × 2.1 mm (Thermo, Waltham, MA, USA) and Analytical: Pursuit 3 C18 Metaguard 10 × 2 mm attached to a Pursuit 3 C18 50 × 2 mm. Samples were eluted from the analytical column using a linear gradient over 20 min. Samples were analyzed using a QTrap 6500+ (Sciex, Framingham, MA, USA) running in positive mode. The ion transitions monitored for TFV-DP were 448–176 and 448/270, and the [13C5]-TFV-DP internal standard ion transitions were 453/181 and 453/275. Calibration curves were generated from standards of TFV-DP spiked into 80% methanol over the range from 0.25 to 10 nM. The lower limit of quantification for TFV-DP was 100 fmol/sample. All calibration curves had r^2^ values greater than 0.99.

### 2.8. Allele-Specific PCR for K65R

Macaques that became infected continued to receive weekly doses of TAF (13.7 or 27.4 mg/kg) for up to three weeks after infection to monitor for risks of drug resistance emergence. The tenofovir resistance mutation K65R was monitored using an allele-specific PCR assay capable of detecting low frequencies of K65R in the RT region, as previously described [34]. 

### 2.9. Data and Statistical Analysis

GraphPad PRISM 7.0 and SAS version 9.4 were used to process and analyze the data. Infection probabilities in treated and placebo controls were compared using Fisher’s exact test. Kaplan–Meier estimates of survival and the log-rank test with a Bonferroni correction were used to compare the survival distribution between animals receiving oral TAF and untreated controls. Hazard ratios were calculated to investigate differences over time between the probability of infection in the treated and untreated groups. Efficacy was calculated by finding the relative risk of infection and subtracting it from 1. All efficacy 95% confidence intervals were calculated using an exact method due to small sample sizes. 

## 3. Results

### 3.1. Selection of Weekly Oral TAF Dose in Macaques

We performed dose-ranging studies in female pigtailed macaques to determine an oral dose of TAF that would maintain TFV-DP levels in PBMCs above 1100 fmols/10^6^ PBMCs for at least one week. The median concentrations of TFV in plasma and TFV-DP in PBMCs are shown in Table 1, and the corresponding concentration-versus-time curves are presented in Figure 1. The 13.7 mg/kg dose of TAF resulted in median (range) TFV-DP levels in PBMCs of 2063 [2025–2530] fmols/10^6^ cells on day 3 and 613 [470–928] fmols/10^6^ cells at day 7 (Figure 1). With the 27.4 mg/kg dose, median TFV-DP levels on day 3 and 6 were 6740 [4410–9070] fmols/10^6^ cells and 4390 [3270–5510] fmols/10^6^ cells, respectively (Figure 1A). We also measured TFV levels in plasma at 5 h, 72 h, and 168 h (Figure 1B). Median peak TFV concentrations ranged between 188 and 284 ng/mL at 5 h in animals dosed with 13.7 and 27.4 mg/kg, respectively, and were below the limit of detection at 72 h (Table 1). 

### 3.2. Efficacy of Weekly Oral TAF (27.4 mg/kg) against Vaginal SHIV Infection

In view of the favorable PK profile of the weekly oral TAF, we first investigated the protection conferred by the 27.4 mg/kg dose in a study design that included once-weekly oral TAF doses followed by twice-weekly vaginal SHIV challenges (days 3 and 6 post-dosing) over six consecutive weeks (Figure 2A). Infection outcomes were compared to six untreated macaques (two real-time and four contemporary controls) also challenged twice-weekly with the same SHIV inoculum under the same experimental conditions. The two real-time controls were both infected at challenge 2. The contemporary controls were infected at challenges 1, 3, 5, and 10. Overall, untreated animals became infected after a median of four challenges (range = 1–10). In contrast, five out of six macaques that received weekly oral TAF doses at 27.4 mg/kg remained uninfected after six weeks of challenges or a total of 12 SHIV exposures (*p* = 0.0087), resulting in a calculated efficacy of 93.9% [95% CI = 61.7–99.8%] (Figure 2B). The TFV-DP levels in PBMCs of the five protected animals ranged between 933 and 17,190 fmols/10^6^ cells throughout the challenge period (Figure 2C). Median TFV-DP levels were 6095 (2250–17,190) and 3290 (933–10,290) at days 3 (open circles) and 6 (closed circles), respectively. Assuming a 7-day eclipse period, the breakthrough infection was estimated to have occurred during or prior to week 2. Notably, this animal had significantly lower TFV-DP in PBMCs (median = 405 [274–677] fmols/10^6^ cells) than the protected animals during the two weeks prior to SHIV RNA detection (*p* = 0.001) (Figure 2C). 

### 3.3. Efficacy of Weekly Oral TAF (13.7 mg/kg) against Vaginal SHIV Infection

Using the same experimental design and procedures, we next investigated if a lower TAF dose of 13.7 mg/kg could also provide a high level of protection against vaginal SHIV infection (Figure 3A). We found that five of the six animals that received weekly oral TAF doses at 13.7 mg/kg remained uninfected after 12 challenges (*p* = 0.0065), resulting in a calculated efficacy of 94.1% [95%CI = 62.9–99.8%] (Figure 3B). The median concentrations of TFV-DP in PBMCs on days 3 (open circles) and 6 (closed circles) in the protected animals were 3110 [83.7–8220] and 1137.5 [584–1861] fmol/10^6^ cells, respectively, compared to 5446 and 1088 fmol/10^6^ cells in the breakthrough infection (Figure 3C).

### 3.4. Efficacy of Weekly Oral TAF (27.4 mg/kg) against Rectal SHIV Infection

Using the same study design and SHIV challenge dose, we next assessed if a higher weekly oral TAF dose of 27.4 mg/kg could also be effective in protecting against rectal SHIV exposures (Figure 4A). Consistent with the six historical controls that were infected after a median of 3.5 challenges (challenge 1, 1, 2, 5, 9, and 9), the two real-time controls were infected at challenge 1 and 5. In contrast, three of the six animals that received weekly oral TAF doses (27.4 mg/kg) remained uninfected after 12 rectal SHIV challenges, resulting in a calculated efficacy of 80.7% [95% CI = 32.8–96.5%]. The detection of SHIV RNA in the breakthrough infections in the other three treated macaques occurred late, with the time of infection estimated at challenges 7, 9, and 12. Moreover, the difference in time to infection between the treated and control groups was statistically significant (*p* = 0.0069, Log-rank test) (Figure 4B). The median concentrations of TFV-DP in PBMCs in protected animals at days 3 and 6 were 4417 [1563–12,833] and 2203 [855–3600] fmol/10^6^ cells, respectively. Interestingly, the median TFV-DP concentrations in animals that became infected were significantly higher on day 3 (9133 [2375–20,367] fmol/10^6^ cells) and 6 (5150 [2607–9367] fmol/10^6^ cells) (*p* = 0.0014 and <0.0001, respectively) (Figure 4C). 

### 3.5. Acute Viremia and Drug Resistance Emergence in Breakthrough Infections

Macaques with breakthrough infections continued to receive weekly oral TAF doses for one to three additional weeks after infection to monitor the effects of TAF exposures on viremia and drug resistance emergence. The peak viremias (log_10_ RNA copies/mL) in the two vaginal breakthrough infections were 3.9 and 4.6, respectively, and were significantly lower than those observed in untreated controls infected vaginally (median = 6.7; *p* = 0.035) (Figure 5A). Although the median peak viremia in the three rectal breakthrough infections was also lower than in untreated controls infected rectally, the difference did not achieve statistical significance (6.4 versus 8.3, respectively; *p* = 0.078) (Figure 5B). These findings are consistent with the potent antiviral activity of TAF and the long intracellular half-life of TFV-DP in PBMCs. None of the five breakthrough infections selected for the K65R mutation were associated with TFV resistance during the additional weeks of TAF dosing or the following drug washout period (Appendix A). 

## 4. Discussion

Simple non-daily PrEP modalities could benefit users who prefer not to use daily oral pills or long acting injectables and thus help expand PrEP coverage. We modeled in macaques if a weekly oral regimen of TAF could be an effective non-daily PrEP modality. We identified a weekly PrEP regimen consisting of oral TAF doses provided at 27.4 or 13.7 mg/kg that maintained TFV-DP levels in PBMCs above 1100 fmols/10^6^ cells for approximately seven days and demonstrated that these levels were highly protective against vaginal SHIV infection (>90% efficacy). However, despite achieving TFV-DP concentrations in PBMCs that ranged between 5000 and 9000 fmol/10^6^ cells, the higher TAF dose (27.4 mg/kg) provided slightly lower protection against rectal SHIV challenges. Our findings point to a clinically achievable weekly oral TAF dose to protect against vaginal HIV acquisition, but the lower rectal efficacy suggests that further dose optimization and a better understanding of the PD correlate is needed for a weekly regimen for rectal protection. 

The weekly oral TAF doses evaluated in our study are likely to be achievable in a clinical setting. Considering it has been shown that a 1.5 mg/kg dose of oral TAF in macaques is equivalent to 25 mg of TAF in humans [34] and that TFV-DP has a similar intracellular half-life in macaques and humans (~5 days) [38], we estimate that the weekly doses of 13.7 and 27.4 mg/kg tested in macaques may be equivalent to approximately 228 and 450 mg of TAF in humans, assuming dose proportionality. However, further clinical PK-assessments are needed to confirm these estimations. It is also worth noting that plasma TFV exposures are transient and that the Cmax values of the weekly doses fall within the range of those achieved with daily doses of 300 mg of TDF in humans [29]. Despite the higher TAF dose, a less frequent dosing regimen with weekly single-agent TAF may still be safer with less renal toxicity than observed with daily oral TDF (300 mg). Clinical studies are needed to understand the safety profile of weekly oral TAF.

In a recent report, a biodegradable implant delivering 0.7 mg of TAF per day yielded sustained and high TFV-DP levels in PBMCs (above 1100 fmol/10^6^ cells), which conferred complete protection against repeated vaginal SHIV exposures. However, this implant resulted in local toxicity and necrosis at the implantation site [31]. The poor tolerability of TAF implants has also been observed with other implant delivery platforms, highlighting the challenges of subcutaneous delivery of TAF released from a reservoir platform [39]. Our efficacy results reaffirm the established vaginal protection threshold for TAF seen with implants and provides an improved modality to safely deliver TAF at effective levels for long-lasting protection.

The reasons for the lower protection against rectal exposures compared to vaginal exposures are not known. In humans and macaques treated with the clinical oral TAF dose (25 mg or 1.5 mg/kg, respectively), TVF-DP concentrations in both rectal and vaginal tissues are generally low or undetectable [40]. In an earlier study, oral PrEP with 13.7 mg/kg of TAF showed no protection against rectal SHIV infection despite achieving high TFV-DP levels in PBMCs [36]. One proposed explanation for the lack of protection was related to reduced drug activity in the rectal mucosa due to high levels of competing natural substrate. TFV-DP is known to compete with the natural substrate dATP during reverse transcription. Because rectal lymphocytes express high levels of endogenous dATP, it is plausible that the high dATP to TFV-DP ratio rendered TFV-DP less effective in blocking infection [36,41]. The high dATP concentrations in rectal lymphocytes are believed to be attributed to the high frequency of activated lymphocytes in the rectum, which may increase dNTP pools [41,42,43]. We show here that increasing the oral TAF dose to 27.4 mg/kg resulted in higher rectal protection, suggesting a potential shift in the dATP/TFV-DP balance that was sufficient to restore antiviral activity.

The analysis of TAF breakthrough infections led to several important observations. Acute viremias in all TAF-treated macaques were blunted despite receiving only one to three additional weekly doses of TAF post-infection. These low viremias likely reflect the potent antiviral activity of TFV-DP and its long intracellular half-life. The rapid decrease in viremia could also explain why none of the animals selected the K65R mutation associated with TFV resistance. These data also suggest that weekly oral TAF may be a good candidate drug to combine with other ARVs for weekly treatment regimens, as interest in developing weekly treatment options continues to expand [44].

Our study has several limitations. We did not measure TFV-DP or dATP levels in vaginal or rectal tissues in an effort not to compromise the mucosal epithelium during the virus challenges. Therefore, we could not establish a direct relationship between protective efficacy and tissue TFV-DP or dATP levels. We used the same SHIV dose for rectal and vaginal challenges as opposed to the typically 5-fold lower SHIV dose generally used for rectal SHIV exposures in the repeat low-dose model [45]. However, the median number of challenges needed to infect untreated animals was similar in both routes, suggesting little or no impact of this dose adjustment on infection outcome. Scaling the weekly macaque TAF dose to humans assumes proportionality of intracellular TFV-DP within an 18-fold range and will require dose-ranging studies in humans to more precisely define the weekly TAF dose. Lastly, we did not monitor systemic toxicities associated with these weekly TAF doses. In a Phase 1b clinical trial in humans, no serious adverse events were reported with daily oral TAF doses of 200 and 600 mg, although this study was limited by the small number of participants that received a single weekly dose.

## 5. Conclusions

In summary, weekly dosing with oral TAF demonstrated high efficacy against vaginal and rectal SHIV infection in two macaque models of HIV PrEP. These preclinical data inform clinical studies of non-daily dosing options that can provide greater flexibility in meeting users’ needs to achieve increased PrEP uptake and adherence. Our findings support the clinical development of weekly oral TAF for long-acting non-daily PrEP. 

## Figures and Tables

**Figure 1 pharmaceutics-16-00384-f001:**
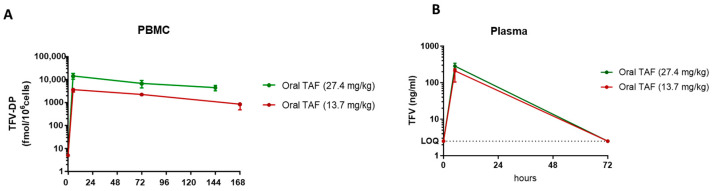
Drug concentration at 5 h, 72 h, 144 h and 168 h after single oral TAF dose. (**A**) TFV-DP concentration in PBMC. (**B**) TFV concentration in plasma. Values reflect median levels ± standard deviation.

**Figure 2 pharmaceutics-16-00384-f002:**
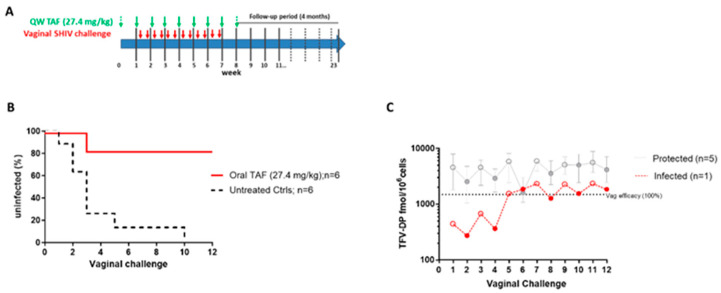
Efficacy of QW oral TAF (27.4 mg/kg) against vaginal SHIV infection. (**A**) Study design. (**B**) Survival curve indicating the cumulative percentage of uninfected macaques as a function of the number of vaginal; SHIV exposures. Time to infection was delayed in macaques treated weekly with oral TAF (*p* = 0.0087). (**C**) TFV-DP levels during vaginal SHIV exposure at day 3 (open circles) and day 6 (closed circles).

**Figure 3 pharmaceutics-16-00384-f003:**
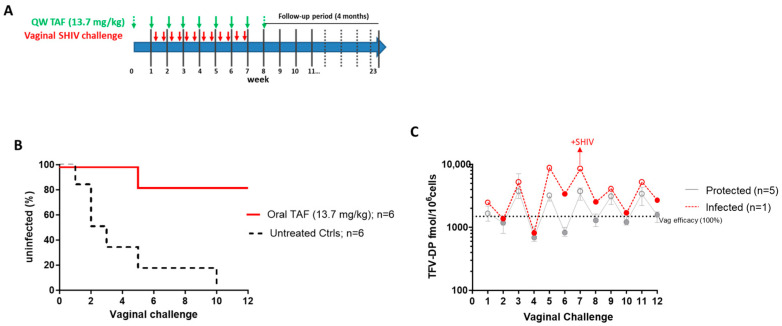
Efficacy of QW oral TAF (13.7 mg/kg) against vaginal SHIV infection. (**A**) Study design. (**B**) Survival curve indicating the cumulative percentage of uninfected macaques as a function of the number of vaginal SHIV exposures. Time to infection was delayed in macaques treated weekly with oral TAF (*p* = 0.0065). (**C**) TFV-DP during vaginal SHIV exposure at day 3 (open circles) and day 6 (closed circles).

**Figure 4 pharmaceutics-16-00384-f004:**
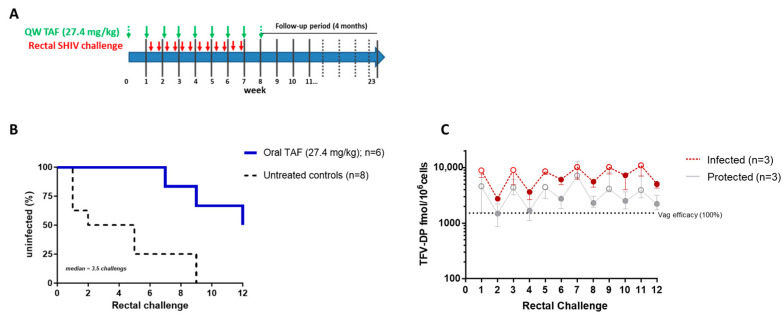
Efficacy of QW oral TAF (27.4 mg/kg) against rectall SHIV infection. (**A**) Study design. (**B**) Survival curve indicating the cumulative percentage of uninfected macaques as a function of the number of rectal SHIV exposures. Time to infection was delayed in macaques treated weekly with oral TAF (*p* = 0.0069). (**C**) TFV-DP during vaginal SHIV exposure at day 3 (open circles) and day 6 (closed circles).

**Figure 5 pharmaceutics-16-00384-f005:**
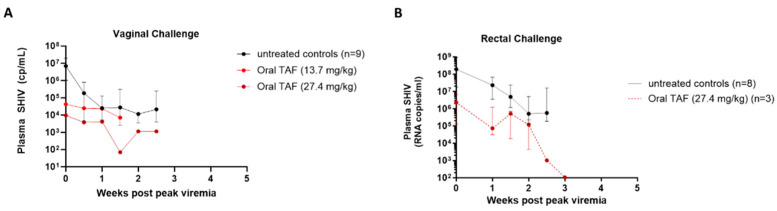
Acute viremia in macaques infected during TAF treatment (**A**) Median peak levels in the breakthrough animals during vaginal challenges following QW TAF dosing at 13.7 and 27.4 mg/kg were significantly lower (*p* = 0.035; log-rank test) (**B**) Median peak viremia levels in the breakthrough animals during rectal challenges were lower than control animals (*p* = 0.078; log-rank test).

**Table 1 pharmaceutics-16-00384-t001:** Pharmacokinetics of oral TAF.

	Plasma			PBMC		
Oral TAF Dose	TFV [T_max_] Hours	TFV [C_max_] ng/mL	TFV [AUC_0–72h_] ng∗h/mL	TFV-DP [T_max_] Hours	TFV-DP [C_max_] fmol/10^6^ Cells	TFV-DP [AUC_0–72h_] fmol∗h/10^6^ Cells
13.7 mg/kg	5	188.2 ± 54.9	6865 ± 1844	5	3552 ± 550.4	203,293 ± 25,735
27.4 mg/kg	5	284.1 ± 139.0	10,319 ± 2714	5	14,090 ± 7700	697,805 ± 391,389

## Data Availability

The data presented in this study are available upon request from the corresponding author.

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
