# Peer review of "Weekly Oral Tenofovir Alafenamide Protects Macaques from Vaginal and Rectal Simian HIV Infection"

_pharmaceutics, 2024, doi:10.3390/pharmaceutics16030384_

Round 1
Reviewer 1 Report
Comments and Suggestions for Authors
In this study, the authors investigated the pharmacokinetics and the efficacy of a once-weekly oral use of tenofovir alafenamide for the protection of vaginal and rectal infection of SHIV in macaques. The study is highly relevant for in the field of HIV prophylaxis. Methods and results presently clearly, with sound discussion and conclusion. I recommend the publication of the manuscript with one minor concern: In both dose groups, the animals with breakthrough infection showed quite different PBMC exposure of TFV-DP. Is there any explanation? And what could the consequence of these results for the intended use of the drug?
Author Response
Reviewer 1
Comments and Suggestions for Authors
In this study, the authors investigated the pharmacokinetics and the efficacy of a once-weekly oral use of tenofovir alafenamide for the protection of vaginal and rectal infection of SHIV in macaques. The study is highly relevant for in the field of HIV prophylaxis. Methods and results presently clearly, with sound discussion and conclusion. I recommend the publication of the manuscript with one minor concern: In both dose groups, the animals with breakthrough infection showed quite different PBMC exposure of TFV-DP. Is there any explanation? what could the consequence of these results for the intended use of the drug?
Thank you for your insightful comments. We agree with the reviewer that this was an unexpected outcome. We can only speculate that the differences in the breakthrough infections may be a result of the following:
The breakthrough infection in the high dose group (27.4 mg/kg) occurred during challenges when TFV-DP levels were below the vaginal protective threshold previously established with TFV-eluting implants (Massud et.al, JAC 2022). The timing of the breakthrough infection suggest that the animal likely became infected due to insufficient drug levels at the time of SHIV exposures.
An explanation of the breakthrough infection in the lower dosage group (13.7 mg/kg), which occurred when TFV-DP levels were similar to those in the protected group, is not clear. Lowering the oral TAF dose may have resulted in lower TFV-DP levels in vaginal tissues, rendering less protection at the point of virus entry. Unfortunately, we were unable to collect tissue samples during the challenge period to measure the ratio of intracellular TFV-DP in PBMCs and tissues.
Our results demonstrate that a weekly oral TAF dose is highly effective in preventing vaginal SHIV infection in a model that predicted the clinical efficacy of daily oral FTC/TDF (Truvada) in women. However, further investigations are needed to evaluate the pharmacokinetics and effectiveness of weekly single-agent TAF in women. Understanding these factors could optimize the intended use of oral TAF for nondaily PrEP in women.
Reviewer 2 Report
Comments and Suggestions for Authors
The manuscript “Weekly oral tenofovir alafenamide protects macaques from vaginal and rectal SHIV infection” aims to support the clinical development of weekly oral tenofovir alafenamide for long acting based on the preclinical studies. The article is well written and easy to follow. However, there are few major aspects which should be addressed prior to publication
1. The methodological details for ion-pair chromatography-MS/MS used in quantifying TFV-DP
2. Elaborate on the procedure for measuring plasma concentrations of TFV.
3. Include the chromatographic data in the supplementary files.
Keeping in view of above observations, I recommend a major revision of the manuscript
Author Response
Reviewer 2
The manuscript “Weekly oral tenofovir alafenamide protects macaques from vaginal and rectal SHIV infection” aims to support the clinical development of weekly oral tenofovir alafenamide for long acting based on the preclinical studies. The article is well written and easy to follow. However, there are few major aspects which should be addressed prior to publication.
Thank you for your comments and suggestions. Please see the answers to your comments below.
- The methodological details for ion-pair chromatography-MS/MS used in quantifying TFV-DP
The methods on ion-pair chromatography-MS/MS used to quantify TFV-DP was previously published (Kuklenyik et al., J Chromatogr B 2009). For clarity of our methods, we now include additional details on pg 3-4 lines 152-160. Briefly, the liquid chromatography method employed the use of a column-switching system (a 6-port valve and a 10-port valve) with 4 single-solvent line pumps, and an autosampler (Shimadzu). The eluents are Pump A: 0.2% acetic acid, Pump B: 1% ammonium hydroxide, Pump C: 10 mM ammonium formate/30% acetonitrile/0.1% dimethylhexylamine (DMHA). The columns used: WAX: Biobasic AX 5um 10x2.1mm (Thermo) and Analytical: Pursuit 3 C18 Metaguard 10x2 mm attached to a Pursuit 3 C18 50x2mm. Samples were eluted from the analytical column using a linear gradient over 20 minutes. Samples were analyzed using a QTrap 6500+ (Sciex) running in positive mode. The ion transitions monitored for TFV-DP were 448-176 and 448/270 and the [13C5]-TFV-DP internal standard ion transitions were 453/181 and 453/275. Calibration curves were generated from standards of TFV-DP spiked into 80% methanol over the range from 0.25 to 10 nM. The lower limit of quantification for TFV-DP was 100 fmol/sample. All calibration curves had r2 values greater than 0.99.
- Elaborate on the procedure for measuring plasma concentrations of TFV.
Additional details on the method and the appropriate reference are now included on pg 3 (lines 142-144). The following description was added in on page 3 (lines 134-144) “Concentrations of tenofovir in plasma were measured by high-performance liquid chromatography-tandem mass spectrometry (HPLC-MS/MS) as previously described (Kuklenyik et al. Chromatogr Sci. 2009). Briefly, a cocktail of 500 µL of methanol containing internal standard, isotopically labeled tenofovir (Toronto Research Chemicals Inc, Toronto, Canada), was added to 100 µL of plasma to precipitate proteins. After a brief centrifugation and removal of protein precipitates, the supernatant was evaporated to near dryness and re-suspended in 100 µL of mobile phase A (0.2% formic acid in water). Ten µL of the final solution was injected onto a UK-C18 column (100 x 1 mm, Imtakt, Portland, OR) connected to an HPLC-MS/MS system. An aqueous-acetonitrile mobile-phase gradient was used to elute the drugs from the column and into the analyzer. Mass transitions used for tenofovir are 288.0/176.3 and 288.0/159.1 (m/z) and were monitored in positive mode. The drug concentrations were estimated from a standard curve with a range of 1-3000 ng/mL using Analyst software 1.7.3 (Sciex, Foster City, CA) and a lower limit of quantification of 10 ng/mL”
- Include the chromatographic data in the supplementary files.
Thank you for the suggestion. However, we believe that providing chromatographic data on the large number of data sets, which includes multiple time points for each animal, would be distractive and not add much value to the paper. The assay for measuring TFV and TFV-DP in macaque specimens have been validated and extensively reported in the literature, including our manuscript recently published in the Journal of Pharmaceutics (Jhunjhunwala et al 2021)
Reviewer 3 Report
Comments and Suggestions for Authors
This is an interesting study that explored the feasibility of administering HIV PrEP once weekly. The study is well-conducted. However, there are a few comments for the authors' consideration.
1. Introduction (line 72): The clinical trial cited is not for tenofovir, but rather for lenacapavir. Please cite the correct clinical trial that reported to serious adverse effects with high-dose TAF.
2. Introduction (line 80-83): This sentence summarizes the study findings and sounds like a conclusion. Please move it to the conclusion on page 7.
3. Methods (2.3 Virus challenge stock): Please specify the concentration of the viral inoculum.
4. Methods: On line 105, you mentioned that only 2 macaques received 27.4 mg/kg, whereas on line 111 you mentioned that they were 6 animals. Please clarify this inconsistency.
5. Methods (lines 114-115): How many hours after the SHIF instillation was the blood sample collected? It's important to allow sufficient time for the virus to distribute and enter the blood of the animal to assess the efficacy of the TAF.
6. Results (lines 169): Add "Mean" to the beginning of the sentence and specify that this range is for both doses since the previous statement concerned the 27.4 mg/kg dose.
7. Results (line 228): A P of 0.078 is not statistically significant. Please edit the sentence to say "numerically lower but did not achieve statistical significance."
8. Discussion: It is worth noting that TAF can adversely affect the lipid profile, which is something to be considered when such high doses of TAF are to be studied in human subjects.
Author Response
Reviewer 3
This is an interesting study that explored the feasibility of administering HIV PrEP once weekly. The study is well-conducted. However, there are a few comments for the authors' consideration.
Thank you for your comments and suggestions. Please see our comments below.
- Introduction (line 72): The clinical trial cited is not for tenofovir, but rather for lenacapavir. Please cite the correct clinical trial that reported to serious adverse effects with high-dose TAF.
We confirm that the clinical trial (NCT03739866) cited is correct. The clinical trial evaluated both LEN and TAF in a two-part study design.
Part A: To evaluate the short-term antiviral activity of Lenacapavir (formerly GS-6207) with respect to the maximum reduction of plasma HIV-1 RNA (log10 copies/mL) from Day 1 through Day 10 compared to placebo in HIV-1 infected adults who are antiretroviral treatment-naive or are experienced but capsid inhibitor (CAI) naive.
Part B: To evaluate the short-term antiviral activity of TAF (200 and 600 mg) with respect to the maximum reduction of plasma HIV-1 RNA (log10 copies/mL) from Day 1 through Day 10 in HIV-1 infected adult subjects who are antiretroviral treatment naïve or are experienced but without resistance to TAF
- Introduction (line 80-83): This sentence summarizes the study findings and sounds like a conclusion. Lines 80-83 were deleted.
- Methods (2.3 Virus challenge stock): Please specify the concentration of the viral inoculum.
The details of the viral inoculum are now included in section 2.3 page 2, lines 92-96. “SHIVS162P3 (SIVmac239 backbone with an HIV-1 subtype B, CCR5–tropic envelope) was obtained from the National Institutes of Health (NIH) AIDS Research and Reference Reagent Program and was propagated in PBMCs from pigtailed macaques to generate a master stock (SHIV162p3 [4,660 TCID50]). The master stock used for vaginal and rectal challenges was diluted to 1:20 and stored individually as 1mL aliquots in liquid nitrogen and thawed prior to each challenge.”
- Methods: On line 105, you mentioned that only 2 macaques received 27.4 mg/kg, whereas on line 111 you mentioned that they were 6 animals. Please clarify this inconsistency.
We clarify that only 2 macaques were used for assessing the PK profile of single dose oral TAF at 27.4 mg/kg. For efficacy assessments, a different group of 6 pigtailed macaques received weekly oral TAF at 27.4 mg/kg.
- Methods (lines 114-115): How many hours after the SHIF instillation was the blood sample collected? It's important to allow sufficient time for the virus to distribute and enter the blood of the animal to assess the efficacy of the TAF.
Blood was collected 30 minutes prior to each SHIV challenge. We clarify that our intention was to understand the amount of drug present around the time of virus exposure as well as determine the SHIV infection status from the previous challenges. We feel that our blood collection schedule appropriately helps to address these questions. Specification on the timing of blood collection is now included on line 127.
- Results (lines 169): Add "Mean" to the beginning of the sentence and specify that this range is for both doses since the previous statement concerned the 27.4 mg/kg dose.
We have now revised lines 188-189 to include the following “Median TFV concentrations ranged between 188 and 284 ng/ml at 5h in the animals dosed with 13.7 mg/kg and 27.4 mg/kg respectively.
- Results (line 228): A P of 0.078 is not statistically significant. Please edit the sentence to say "numerically lower but did not achieve statistical significance."
Thank you for pointing this out. Correction included (pg 6 line 249-250)
- Discussion: It is worth noting that TAF can adversely affect the lipid profile, which is something to be considered when such high doses of TAF are to be studied in human subjects.
We agree and believe that clinical studies are needed to understand the safety profile of weekly dosing with single-agent TAF on lipid profiles. We now include this language on page 7 line 278-279
Round 2
Reviewer 2 Report
Comments and Suggestions for Authors
The authors have addressed all the raised comments, and the manuscript can be accepted in its current form.